# Biotic and Abiotic Elicitors of Stilbenes Production in *Vitis vinifera* L. Cell Culture

**DOI:** 10.3390/plants10030490

**Published:** 2021-03-05

**Authors:** Martin Sák, Ivana Dokupilová, Šarlota Kaňuková, Michaela Mrkvová, Daniel Mihálik, Pavol Hauptvogel, Ján Kraic

**Affiliations:** 1Department of Biotechnology, Faculty of Natural Sciences, University of Ss. Cyril and Methodius, Námestie J. Herdu 2, 91701 Trnava, Slovakia; sak.martin@liqoil.sk (M.S.); sarlota.kanukova@gmail.com (Š.K.); daniel.mihalik@ucm.sk (D.M.); 2Liqoil-Pharm s.r.o., Bratislavská 535, 90046 Most pri Bratislave, Slovakia; dokupilova.ivana@liqoil.sk; 3Department of Biology, Faculty of Natural Sciences, University of Ss. Cyril and Methodius, Námestie J. Herdu 2, 91701 Trnava, Slovakia; michaela.mrkvova@ucm.sk; 4Research Institute of Plant Production, National Agricultural and Food Centre, Bratislavská cesta 122, 92101 Piešťany, Slovakia; pavol.hauptvogel@nppc.sk

**Keywords:** *Vitis vinifera*, cell culture, elicitation, fungal elicitors, stilbenes

## Abstract

The in vitro cell cultures derived from the grapevine (*Vitis vinifera* L.) have been used for the production of stilbenes treated with different biotic and abiotic elicitors. The red-grape cultivar Váh has been elicited by natural cellulose from *Trichoderma viride*, the cell wall homogenate from *Fusarium oxysporum* and synthetic jasmonates. The sodium-orthovanadate, known as an inhibitor of hypersensitive necrotic response in treated plant cells able to enhance production and release of secondary metabolite into the cultivation medium, was used as an abiotic elicitor. Growth of cells and the content of phenolic compounds *trans*-resveratrol, *trans*-piceid, δ-viniferin, and ɛ-viniferin, were analyzed in grapevine cells treated by individual elicitors. The highest accumulation of analyzed individual stilbenes, except of trans-piceid has been observed after treatment with the cell wall homogenate from *F. oxysporum*. Maximum production of trans-resveratrol, δ- and ɛ-viniferins was triggered by treatment with cellulase from *T. viride*. The accumulation of trans-piceid in cell cultures elicited by this cellulase revealed exactly the opposite effect, with almost three times higher production of trans-resveratrol than that of *trans*-piceid. This study suggested that both used fungal elicitors can enhance production more effectively than commonly used jasmonates.

## 1. Introduction

Plants produce secondary metabolites with different and specific functions. They operate as antioxidants, enzyme inhibitors, signal molecules and growth regulators, but have also other roles. Some secondary metabolites synthesize plants after the attack of microorganisms as biocidal substances. Their synthesis is a response to the infection itself and they also inhibit the growth of invading pathogens. Phytoalexins are one of the best and longest studied groups of secondary metabolites belonging into a group of the low molecular weight compounds [1]. The class of phytoalexins isolated from the grapevine (*Vitis vinifera* L.), the oligomeric forms of the stilbene resveratrol, have been termed viniferins [2]. Stilbene phytoalexins constitute specific group of molecules containing the skeleton of trans-resveratrol. They occur only in a limited number of families including the Vitaceae. In addition to resveratrol, other stilbenes found in grapevine are ε-viniferin, α-viniferin, *trans*-pterostilbene, *trans*- and *cis*-piceid, trans- and *cis*-astringin, and trans- and cis-resveratrol-oside [3,4]. The grapevine stilbenes are increasingly studied due to their beneficial biological activities. The best known is resveratrol. It can be ingested daily in food, therefore, a number of dietary effects related to resveratrol have been reviewed [5]. Resveratrol and its derivatives have antibacterial, antifungal, antioxidant, anticancer, cardioprotective, neuroprotective and antiaging effects, as well as other biological activities [6,7,8,9,10]. Significant antioxidant activity and cytoprotective effects of thirteen different stilbenes, including the most abundant ε-viniferin and resveratrol, were also presented [11]. The properties of stilbenes have initiated a great demand for their production and applications in nutraceutical, pharmaceutical and cosmetic products.

Stilbenes, as well as resveratrol itself, can be produced in several ways. Extractions from natural plant sources are bounded due to limited plant sources and their low concentration in plant tissues. The complexity of the stilbene biosynthetic pathway and low yield does not favor their production by chemical synthesis. The promising scalable production strategy should be the biotechnological approach, especially using the genetically engineered microorganisms, plants and animal cell lines [12]. Three in vitro cultivation systems (the cell suspension cultures, callus cultures and hairy root cultures) were developed and continually improved for production of stilbenes [13]. However, the in vitro induction, regulation of stilbenes biosynthesis and the accumulation of *trans*-resveratrol, *trans*/*cis*-piceid, ε-viniferin, δ-viniferin, pterostilbene, and *trans*-astringin are mainly associated with the use of cell suspension cultures [14,15,16,17]. Biotechnology approaches of in vitro production of resveratrol and other stilbenes have already been reported in scale up, using 2–20 L bioreactors [18,19,20,21]. The considerable advantage of such plant-cell systems is the possibility to enhance the yield of required compounds by the application of elicitors [14,19,22] or genetic transformations, for example, using the Agrobacterium gene *rolB* [23]. Generally, the high-added value bioactive compounds can be produced biotechnologically using plant cell factories [24].

Applications of suitable elicitors and combinations of different stress stimuli can enhance the production of desired secondary metabolites by synergistic effect [25]. Elicitors may be classified as biotic with defined or complex composition, abiotic physical or chemical, as well as exogenous or endogenous [26,27,28], respectively. Elicitations of cells in cultures in vitro of different cultivars within the Vitis species usually enhance the accumulation of stilbenes in comparison with untreated cells. Elicitors belonging to the lipid-derived family of jasmonates are naturally activated in plants as a response to stress [29]. The more active derivative methyl jasmonate (MeJA) and the less active cis-jasmonate and dihydrojasmonic acid are main plant stress compounds involved in the signaling of defense responses [30], developmental processes [29,31] and the induction of genes involved in phytoalexin biosynthesis and production of phenolics [32]. The MeJa is very frequently used elicitor applied in different experiments including in vitro systems, as well as in assays of gene expression of defense molecules in Vitis [33,34]. Jasmonates are responsible for the accumulation of secondary metabolites in cell suspension cultures derived from different cultivars of *Vitis vinifera* L. [35,36]. On the contrary, not well-known elicitation function has the abiotic compound sodium orthovanadate. It was able to inhibit the hypersensitive necrotic response of plant cells triggered by bacteria [37] and allow the increase of production and the release of the *cis*-resveratrol into the cultivation medium from elicited grapevine cells [38]. The group of complex biotic elicitors contains components from fungi or yeasts. They are able to induce a hypersensitive response in plants to prevent spreading of infection generally characterized by the cell plasmolysis and cell browning leading to the cell death. The grapevine cells in liquid culture treated with cellulase from *Trichoderma viride* also showed hypersensitive-like response associated with the formation of stress metabolite resveratrol as well as with peroxidase and products of resveratrol oxidation [39,40,41]. The application of filtrate from cultivation media of *Trichoderma atroviride* also induced the hypersensitive-like response of grapevine cells, leaded to decreasing of the cell viability and also to up-regulation of genes involved in stilbenes biosynthesis [42]. Then, carefully adjusted amount and concentration of components from fungi such as *Aspergillus niger*, *Fusarium moniliforme*, *Pheomoniella chlamydospora*, *Trichoderma viride*, *Pseudomonas syringae* can be used for elicitation of secondary metabolism of phytoalexins in cell, callus and root cultures of grapevine as well as in other plant species [27,43,44]. The *Fusarium oxysporum* is also interestingly considered as one of many factors [45] or even as the main factor [46] associated with the decline and death of grapevine plants. The main phytotoxic metabolite produced by *F. oxysporum* isolated from diseased declined grapevine plants was identified as fusaric acid [47]. Extract from sonicated pellet of *F. oxysporum* as well as supernatant were used for elicitation of callus cultures of grapevine, but production of resveratrol was lower in comparison with jasmonic acid used as elicitors. However, fungi also contain another oligosaccharides, proteins and polyunsaturated fatty acids that can trigger the self-defense hypersensitivity reaction and elicit production of phytoalexins [48]. Another defined or complex biotic elicitors used for production of secondary metabolites in grapevine callus and shoot cultures were filtrates, centrifugates, or isolated polysaccharides from yeast, algae and plants [18,49,50].

The aim of this study was to study the competence of red-grapevine (*Vitis vinifera* L.) cultivar Váh to produce secondary metabolites stilbenes in cell suspension cultures and to determine the competence of several less used biotic and abiotic elicitors to increase stilbene production in vitro.

## 2. Results

The production of different stilbene derivatives has been initiated in cell cultures of the red-grapevine cultivar Váh. The enhancement of production was induced by biotic elicitors, methyljasmonate (MeJA), *cis*-jasmonate (cJA), dihydrojasmonic acid (2HJA), cellulase from *Trichoderma viride* (TvC), cell wall homogenate from fungus *Fusarium oxysporum* (FOE) as well as with abiotic elicitor the sodium orthovanadate (NaVO). Many of these compounds are also inducers of defense mechanisms, particularly of the hypersensitive response in the grapevine in vitro cell culture [39,44,51]. All applied elicitors affected the growth and viability of cells in culture, but also their ability to produce stilbenes *trans*-resveratrol, *trans*-piceid, ε-viniferin and δ-viniferin. Generally, the application of elicitors caused the color changes of the cell cultures from green to brown, due to the endogenously increased overproduction of hydrogen peroxide after elicitation [38]. Browning of cells is accompanied by the activation of the oxidative phenolic metabolism [39].

### 2.1. Cell Biomass Growth

The growth and dry weight of cultivated grapevine cells have been influenced by all applied elicitors and were associated with color changes to brown, similarly to our previous experiments with MeJA [52]. Both elicited and non-elicited cell cultures increased the growth throughout the cultivation, up to the final 13th day (i.e., 7 days after elicitation. The dry weight of cells elicited with MeJA1, MeJA2, 2HJA, cJA, and FOE was statistically significantly (*p* ≤ 0.05) higher as early as in 8th day of cultivation, that is, 2 days after treatment with elicitor (Figure 1). The dry weight of TvC1 and NaVO3-induced cells was statistically significantlly (*p* < 0.05) higher only on the last day of cultivation. Treatment with TvC2 did not enhanced this parameter statistically significantly. The dry weight of cells treated by all used elicitors was statistically significantly (*p* < 0.05) lower than in control cells. The non-elicited (control) cell cultures increased cell growth from 5.9 g/L to 8.1 g/L. Cells in cultures treated individually with six used elicitors grew weaker in comparison with the control culture. They increased growth from the same starting value 5.9 g/L to maximum 6.8 g/L (elicited by cJA). Three jasmonates (MeJA, 2HJA, cJA), FOE, TvC1, TvC2, and NaVO affected the cell growth essentially in the same way. Dry weights of cells correlated largely with the cell viability determined by the 2,3,5-triphenyl tetrazolium test [53] adopted for grapevine cell culture [42]. Viability of cells in culture decreased over time. In the beginning, that is, on the 6th day post sub-culture to new media was 90%. Viability decreased to 86% in the untreated control on the day 13. After elicitation with MeJA1, MeJA2, cJA, 2HJA viability decreased to 84–79%. The FOE reduced the viability to 81%, TvC1 and TvC2 to 77% and 73%, respectively. Treatment with NaVO reduced cell viability the most intensively—to only 61%.

### 2.2. Production of Stilbenes

The presence of stilbenes, including trans-resveratrol, trans-piceid, dimeric ε- and δ-viniferin in cell cultures was confirmed by the High Performance Liquid Chromatography (HPLC) analysis. The target compounds were identified according to relative retention time, followed by comparison of the UV-spectra against the authentic standards (Figure 2). Quantitative differences were assessed after calibration using the analytical standards and concentrations were calculated by linear regression.

To increase the synthesis of stilbenes in vitro, the grapevine cells were treated with biotic and abiotic elicitors. Content of stilbenes was determined on the 8th, 11th, and 13th day of cultivation, that is, 2nd, 5th, and 7th day after the treatment by elicitors. The production of stilbenes has been affected by all used elicitors. Generally, the highest enhancement of target stilbenes production, with the exception of trans-piceid, has been detected in cells treated with cellulase from *Trichoderma viride* (TvC1, TvC2 variants, Table 1). The level of trans-resveratrol at the end of cultivation was 39 times higher in cells elicited by TvC2 in comparison with untreated control cells. Lower concentrations of this elicitor (treatment TvC1) and higher concentrations of MeJA (treatment MeJA2) also effectively increased (14–19 times) production of trans-resveratrol. The production of the glycosylated form of trans-resveratrol, the trans-piceid, has been the most effectively enhanced (2.5 times) by treatment with the cell wall homogenate from fungus *Fusarium oxysporum* (FOE). The most effective elicitors of both the viniferins was TvC. The TvC2 treatment enhanced synthesis of ε-viniferin by 53 times and δ-viniferin almost 10 times in comparison with non-elicited cells.

The effects of all used elicitors on production of trans-resveratrol and trans-piceid during cultivation are presented in Figure 3. The level of produced stilbenes has mostly been increased from the time the elicitor was added up to final 13th day of cultivation in all used elicitors. Elicitors NaVO and FOE, which were reported previously, were only rarely very effective in the improvement of trans-piceid production. On the contrary, neither the cJA nor the 2HJA were effective in the elicitation of trans-resveratrol. Quantitative shifts were observed during the cultivation, both in control and in elicited cell cultures. The decrease of trans-resveratrol in the control culture occurred on the 11th day of cultivation and then increased. No correlation between the production of trans-resveratrol and trans-piceid could be detected. Some elicitors (MeJA2, TvC1, TvC2) triggered the production of trans-resveratrol better than that of its glycoside.

The δ-viniferin is a resveratrol dehydrodimer and an isomer of the ε-viniferin. This compound has been reported as a molecule produced in vitro by the oxidative dimerization of resveratrol by plant peroxidases or fungal laccases and was identified in grapevine cell cultures as well as in wines [54]. The effects of all used elicitors on the production of δ-viniferin and ε-viniferin during cultivation are presented in Figure 4. The production of these two stilbenes increased from the time of addition of the elicitor to the 13th day of elicitation in most of the elicitors used. The opposite occurred when cells were treated with cJA and 2HJA. Elicitations by NaVO and especially by FOE not affected synthesis of both viniferins as effectively as elicited trans-resveratrol and trans-piceid. Both the cJA and 2HJA have been proven as not effective in the elicitation of δ-and ε-viniferin.

## 3. Discussion

The effectiveness of less frequently used biotic substances derived from *Trichoderma viride*, *Fusarium oxysporum*, and the abiotic compound sodium orthovanadate were tested and compared with that of commonly applied and well described three jasmonate elicitors (MeJA, cJA, 2HJA). Exogenous jasmonic acid, its methyl ester and other derivatives, as well as elicitors derived from fungi can regulate growth, promote senescence, and specifically alter the gene expression in plants [48,55]. The level of suppression of the basic physiological processes, such as cell growth, can correlate with the level of secondary metabolites also in elicited grapevine cell cultures. This indicates that elicitors stimulate the plant defense and activate the secondary metabolism in detriment of the basic cellular processes as the primary metabolism, cell division and cell growth [56,57]. A greater negative impact on the growth of grapevine cells in suspension culture had elicitation with the cellulase from *Trichoderma viridae* in both used concentrations and with the abiotic sodium orthovanadate.

The production of trans-resveratrol and *trans*-piceid were found to be differently affected by used elicitors; however, the cellulase from *Trichoderma viridae* in higher concentrations (variant TvC2 in this study) was the most effective. Generally, the fungi of the genus *Trichoderma* possess great potential for the induction of plant resistance against biotic and abiotic stress factors. Due to this they can be used as ecologically friendly bio-control agents for agriculture and sustainable environmental management [58]. Non-elicited grapevine cells accumulated stilbenoid glucoside *trans*-piceid in relatively high amount in comparison with the corresponding aglycone trans-resveratrol. The same occurred in cells elicited with MeJA1, cJA, 2HJA, NaVO and FOE. This suggests that there is a correlation with natural conditions when plants modify secondary metabolites, including stilbenes, by glycosylation and the piceid content is usually higher than the resveratrol content [59]. Glycosylation of stilbenes can be involved in the process of their storage, transport from cytoplasm to apoplasm and protection from peroxidative degradation. Plant cells glycosylate stilbenes by endogenous non-specific glucosyl-transferase and subsequently deposit them in vacuoles to protect itself against their potentially toxic effects [60,61]. The production of *trans*-resveratrol in grapevine cells after treatment with cellulase from *Trichoderma viridae* has also been described previously [39]. A significant increase of the endogenous level of hydrogen peroxide caused by treatment with cellulase from *Trichoderma viridae* accompanied the activation of the oxidative phenolic metabolism and cell culture browning. It has been observed that these cell cultures produced a higher amount of metabolites of the phenylpropanoid pathway [62,63]. Overproduction of trans-resveratrol in cell cultures treated with *T. viride* cellulase in both used concentrations in our study (TvC1, TvC2) can be explained by the effect of cellulases. This was reported also in experiments reaching high content of resveratrol by the biotransformation of piceid with cellulases in *P. cupsidatum* [59]. A similar production of *trans*-resveratrol (about 200 μg/g of DW) has been reported in Vitis cell cultures elicited by centrifugate of the fungus *Aerobasidium pullulans* [50].

The homogenate from fungi *Fusarium oxysporum* (FOE) provided the most significant increasing of *trans*-piceid production in comparison with control. Both the trans-resveratrol and *trans*-piceid can contribute to the antioxidative reactions. Previous studies showed that the *trans*-piceid alone or in combination with *trans*-resveratrol has many associations with inflammation and oxidation by regulating of oxygen species (ROS). *Trans*-piceid also regulates oxidative stress by production of extracellular nitric oxid, mitochondrial superoxide anions and scavenger enzymes in vitro [64,65]. This can explain higher *trans*-piceid production after elicitation leading to the generation of ROS.

Immediate production of *trans*-resveratrol accompanied by browning of cells has been found in cells treated with sodium orthovanadate (NaVO); however, the level of production was not sufficient.

Viniferins as the products of oligomerization resulted from oxidative coupling catalysed by peroxidase isoenzymes are localized in a vacuole, cell wall, and apoplast of grapevine cells [66]. In fact, the elicitation of grapevine cell cultures leads to changes in isoenzyme patterns of extracellular peroxidases, which are probably responsible for the elicitor-induced accumulation of resveratrol oligomers [40]. The *trans*-resveratrol is also a key precursor of different oligostilbene dimers (ɛ-viniferin), trimers (α-viniferin), and tetramers (R-viniferin and R2-viniferin). Two main dimers, the δ- and ɛ-viniferins, were detected in all treated cells in our study, but the ɛ-viniferin was not produced after treatment with the cJA and 2HJA. On the other hand, the δ-viniferin was produced in cells elicited by the cJA and 2HJA, but only in minimum amounts. The maximum production of ɛ-viniferin was achieved in cells treated with TvC2 (3267.6 μg/g DW) and TvC1 (3151.3 μg/g DW), similarly to production of δ-viniferin treated with TvC2 (4587.2 μg/g DW), TvC1 (4486.2 μg/g DW). It might be speculated that a large part of trans-resveratrol was quickly transformed into ɛ- and δ-viniferins. Other reports [50] presented lower production of viniferins in grapevine calluses (550 μg/g DW) and also in cell cultures [35]. It should be emphasized that different genotypes (cultivars) of grapevine were used in different experiments moreover, the regulation system of the stilbene biosynthetic pathway is cultivar specific in *V. vinifera* [67]. The amplitude of the response to elicitation strongly differs between grapevine genotypes. Genetic variation in stilbene accumulation relates to different transcription of several genes such as chalcone synthase and stilbene-related genes the phenylalanine ammonium lyase, stilbene synthase and resveratrol synthase [68,69].

The sodium orthovanadate is reduced by cytoplasmic glutathion and inhibits Na+, K+-ATPase, acid- and alkaline phosphatase, adenyl kinase, as well as several enzymes in the glycolytic pathway. It can adopt a stable trigonal bipyramidal structure and is able to replace phosphate as a substrate [70]. The NaVO was described as a stimulant for resveratrol production in *V. vinifera* cell suspension cultures [38]. In contrast to this, NaVO has not significantly affected production of resveratrol in our experiments. Also decreasing of the cell growth by NaVO was not dramatically different from other used elicitors as was presented previously in cell growth in grapevine callus cultures [20].

Bioproduction in vitro of stilbenes should be improved by combination of biotic and abiotic elicitors and by combinations of elicitors with other chemical compounds such as carbohydrates [71] cyclodextrins, phytotoxins [56,72], ethephon [73], β-glucan, amberlite [18] or physical factors such as ultrasound [74], UV-C irradiation [75]. Future progress in the enhancement of stilbenes production in elicited grapevine cell in vitro will probably be based on genetically improved genotypes of grapevine with engineered genetic background for stilbenes synthetic pathway. Such grapevine can overexpress transgene for stilbene synthase and production of synthesized trans-resveratrol should be dramatically improved in comparison with non-transgenic grapevine [76,77,78]. Cell cultures derived from metabolically engineered grapevine produced also other biologically active analogues of resveratrol, the pterostilbene and piceatannol [79]. Another way for the efficient bio-production of resveratrol and other stilbene derivatives are genetically tailored bacteria and yeasts [13].

## 4. Materials and Methods

### 4.1. Reagents and Standards

Reagents of analytical grade used for HPLC analyses included the standards *trans*-piceid, ɛ-viniferin, and *trans*-resveratrol (Sigma-Aldrich, St. Louis, MI, USA). Other used reagents were methyljasmonate (MeJA), *cis*-jasmonate (cJA), dihydrojasmonic acid (2HJA), cellulase from *Trichoderma viride* (TvC), sodium orthovanadate (NaVO) (Sigma-Aldrich, St. Louis, MI, USA), naphthaleneacetic acid (NAA), 6-benzylaminopurine (BAP), sucrose, agar, yeast extract, bacto-peptone, magnesium sulphate heptahydrate, potassium dihydrogen phosphate, sodium dodecylsulphate (SDS) (Sigma-Aldrich, St. Louis, MI, USA; Duchefa BIOCHEMIE B.V, Netherlands; Imuna, Slovakia, Merck, Germany). Water was purified by the Direct-Q^®^ 3 UV Water Purification System (Merck Millipore, Darmstadt, Germany). Reference standard solutions of polyphenols were prepared in mixture methanol:water (1:1, *v*/*v*) and stored at 4 °C in dark.

### 4.2. Cell Suspension Cultures

Callus cultures were initiated from surface sterilized leaf segments of grapevine (*V. vinifera* L.) cultivar Váh producing blue berries of the “Cabernet” type. It originated from the crossing of the French varieties Castets × Abouriou noir and has been registered in Slovakia in the year 2011. Leaf segments were cultivated on the Murashige & Skoog [80] medium contained salt reduced to one half (½ MS), 3% of sucrose, 0.7% (*w*/*v*) of agar, 0.1 mg/L of NAA, and 0.2 mg/L of BAP, pH 5.8. Explants were cultivated at 25 ± 1 °C under the photoperiod of 16 h light/ 8 h darkness. The cell cultures were initiated from calli in 250 mL Erlenmeyer flasks containing 80 mL of ½ MS liquid medium and cultivated with shaking (120 rpm) under the same physical conditions as the callus cultures. Cell cultures were transferred to a fresh cultivation medium every 30 days. Growth of cultivated cells was monitored and analyzed for 13 days and subsequently established by a comparison of dry weight and increased content of phenolic compounds of treated and control cells, respectively. Volume of 3–4 mL of the cell suspension from each flask was centrifuged at 8000 rpm for 10 min at 10 °C and the deeply frozen pellet was lyophilized and finally weighed.

### 4.3. Elicitor Preparation and Elicitor Treatment

The homogenate of fungal cell walls from cultivated *Fusarium oxysporum* (FOE) was prepared from the cortex of grapevine plant infected by this fungus and further maintained on the potato-dextrose agar (PDA) plates. After five days, the PDA culture was transferred into a 500 mL flask containing 150 mL of the Czapek Dox Broth, 0.5% of yeast extract, and 0.5% of malt extract. FOE containing polysaccharide and protein components has been prepared according to [81,82]. Concentration of polysaccharide was determined by the phenol-sulphuric acid method [83] and concentration of proteins using the BCA Protein Assay kit (Thermo Fisher Scientific, Waltham, MA, USA). Other elicitors, including MeJA, 2HJA, and cJA, were dissolved in 50% of aqueous methanol and methanol aliquots were also added to the control cells. Commercial cellulase with the enzymatic activity of 9 U mg^-1^ obtained from *Trichoderma viride* (TvC) as well as NaVO were dissolved in deionized water. Solutions of elicitors were filtered through 0.22 μm membrane syringe filters and added to the cell suspension culture on the 6th day of cultivation. The final concentrations of elicitors were 200 μM of MeJA (MeJA1), 500 μM of MeJA (MeJA2), 200 μM of 2HJA, 200 μM of cJA, 0.5 U m/L of TvC (TvC1), 1.5 U m/L of TvC (TvC2), 50 mg/L of FOE (carbohydrate equivalent), and 3 mM of NaVO_3_, respectively.

### 4.4. Extraction of Stilbenes

The freeze-dried plant material was ground using the Tissue Lyser II (Qiagen, Germany) and 50 mg of the fine powder was extracted in with 70% aqueous methanol (1:15, *w*/*v*) using sonication in darkness for 1 h. Two independent extractions for each sample were carried out. Insoluble cell debris was removed by centrifugation at 12,000 rpm for 10 min followed by filtration through a lab-made C8 cartridge with particle size 5 μm and pore size 120 Å (Agilent Technologies Inc., Santa Clara, CA, USA). Each sample was rinsed with 200 μL of 70% aqueous methanol in order to obtain the final concentration in range of the calibration curve for the HPLC measurements.

### 4.5. Chromatographic Analysis

Qualitative and quantitative analyses of extracts from cell cultures were carried out by the HPLC system Agilent 1260 Infinity LC with Diode Array Detector (Agilent Technologies Inc., Santa Clara, CA, USA) using a Hypersil BDS C18 column (250 × 4.6 mm, 5 μm particle size) with the following solvent system: Eluent A contained water, tetrabutylammonium hydroxide (40%, *w*/*w* in water), and phosphoric acid (99%) in ratio 99.8:0.1:0.1 (*v*/*v*/*v*), Eluent B contained methanol. The elution gradient had the following profile: 0–5.1 min 20% of B, 5.1–15 min 40% of B, 15–20 min 90% of B, kept for 6 min. The flow rate was 0.8 mL/ min and the column temperature was set to 35 °C. Simultaneous monitoring was performed at 230 nm, 256 nm, 300 nm, and 330 nm, respectively. Relative retention times and UV spectra of peaks were compared to those of the standards for identification. The ferulic acid was used as the internal standard. The compounds were quantified based on the eight points calibration curve of each standard solution in the concentration range of 0.01875, 0.025, 0.0375, 0.05, 0.075, 0.1, 0.15, and 0.2 mg/mL, respectively, prepared from stock solutions. Chromatographic analyses of all four stilbenes were performed in duplicate.

### 4.6. Statistical Evaluation

Experimental data were evaluated by Analysis of Variance (ANOVA) using the software Statistica 7 (StatSoft Inc., Tulsa, OK, USA).

## 5. Conclusions

The newly registered red-grapevine (*Vitis vinifera* L.) cultivar Váh originating from Slovakia seems to be an effective source of cells for the biotechnological production of stilbenes. The highest amount of *trans*-resveratrol and *δ*- and *ε*-viniferins, were recorded in grapevine cell suspension cultures treated with cellulose from *Trichoderma viride*. The highest amount of *trans*-piceid was produced after treatment with cell wall homogenate from *Fusarium oxysporum*. These biotic elicitors were more effective than the commonly used jasmonates and the heavy metal containing sodium orthovanadate.

## Figures and Tables

**Figure 1 plants-10-00490-f001:**
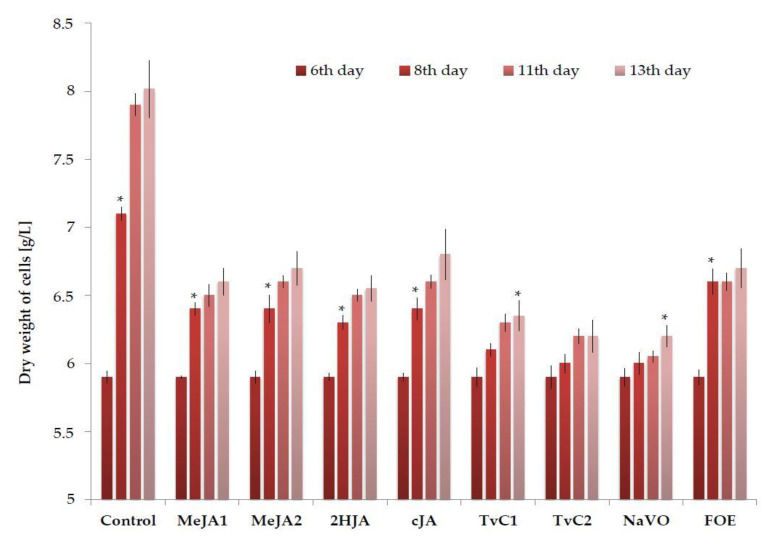
Changes in dry weight of control and elicited grapevine cells during cultivation in vitro on the 6th, 8th, 11th, and 13th day of cultivation (*—statistical significance, *p* ≤ 0.05).

**Figure 2 plants-10-00490-f002:**
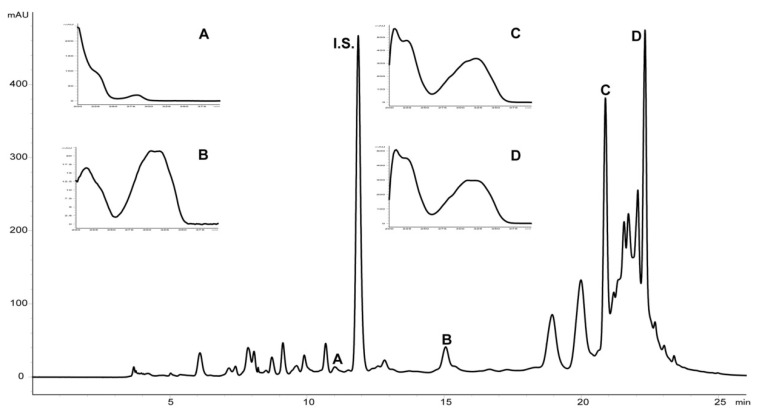
HPLC analysis of extracts from grapevine cell cultures on the 13th day of cultivation, that is, 7 days after treatment with TvC2. A—trans-piceid, B—trans-resveratrol, C—ε-viniferin, D—δ-viniferin, I.S.—internal standard (ferulic acid).

**Figure 3 plants-10-00490-f003:**
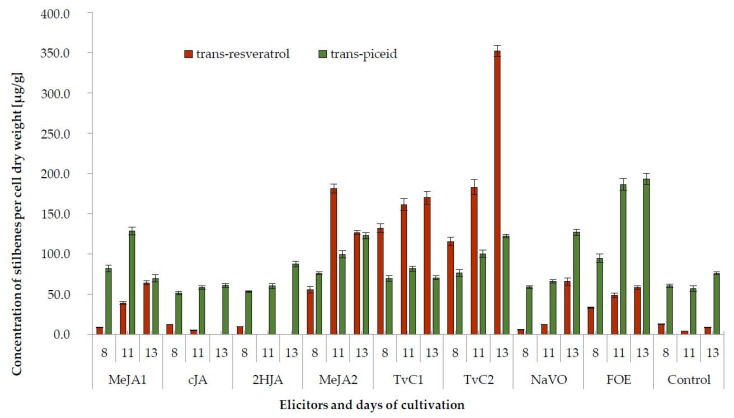
Production of *trans*-resveratrol and *trans*-piceid in control and elicited grapevine cell suspension cultures on the 8th, 11th, and 13th day of cultivation, that is, 2, 5, and 7 days after elicitation, respectively.

**Figure 4 plants-10-00490-f004:**
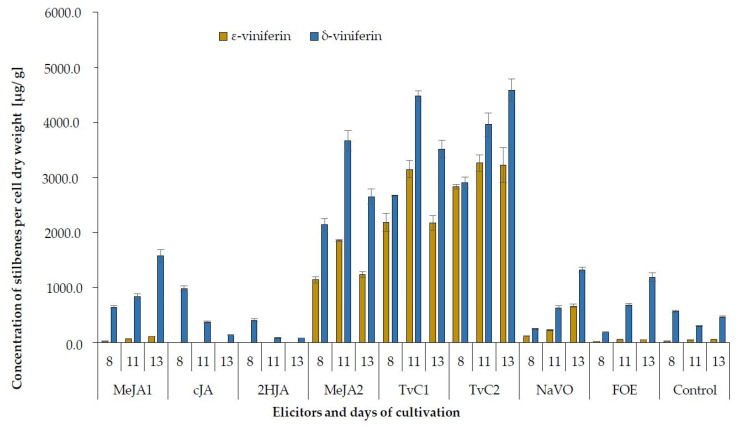
Production of ε-viniferin and δ-viniferin in control and elicited grapevine cell suspension cultures on the 8th, 11th, and 13th day of cultivation, that is, 2, 5, and 7 days after elicitation.

**Table 1 plants-10-00490-t001:** The mean content of stilbenes in control and elicited (on the 6th day) grapevine cells cultured in vitro (μg/g, ^a–g^ statistically significant differences between values, n.d. not detected).

Elicitor	Days of Cultivation	*trans*-Resveratrol	*trans*-Piceid	*ε*-Viniferin	*δ*-Viniferin
MeJA1	8	8.5 ± 0.52 ^ab^	82.0 ± 4.10 ^e^	29.2 ± 1.46 ^a^	647.1 ± 22.42 ^c^
cJA		12.0 ± 0.07 ^b^	51.6 ± 1.81 ^a^	n.d.	985.5 ± 48.61 ^d^
2HJA		9.4 ± 0.14 ^ab^	53.1 ± 0.81 ^a^	n.d.	410.9 ± 22.63 ^b^
MeJA2		55.6 ± 4.17 ^d^	76.1 ± 1.58 ^d^	1141.4 ± 57.45 ^b^	2144.4 ± 113.47 ^e^
TvC1		132.2 ± 5.76 ^f^	69.2 ± 3.48 ^c^	2188.1 ± 167.12 ^c^	2662.9 ± 15.37 ^f^
TvC2		115.6 ± 5.21 ^e^	76.4 ± 4.34 ^d^	2836.5 ± 43.33 ^d^	2907.2 ± 102.10 ^g^
NaVO		5.5 ± 0.22 ^a^	58.9 ± 1.80 ^bc^	121.7 ± 6.25 ^a^	248.3 ± 7.59 ^a^
FOE		32.9 ± 0.95 ^c^	94.7 ± 5.22 ^f^	25.4 ± 0.76 ^a^	193.9 ± 5.13 ^a^
Control		12.8 ± 0.56 ^b^	60.4 ± 2.12 ^ab^	28.6 ± 1.03 ^a^	571.3 ± 20.60 ^c^
MeJA1	11	39.0 ± 1.62 ^c^	128.5 ± 4.86 ^f^	71.5 ± 6.11 ^a^	834.2 ± 55.54 ^d^
cJA		5.2 ± 0.16 ^ab^	58.4 ± 2.21 ^a^	n.d.	376.3 ± 17.38 ^b^
2HJA		n.d.	60.3 ± 3.04 ^a^	n.d.	85.6 ± 4.71 ^a^
MeJA2		181.3 ± 5.44 ^f^	99.6 ± 4.60 ^e^	1859.6 ± 18.60 ^c^	3670.3 ± 184.73 ^e^
TvC1		161.5 ± 7.28 ^e^	81.7 ± 3.30 ^c^	3151.3 ± 157.57 ^d^	4486.2 ± 89.72 ^f^
TvC2		183.0 ± 9.15 ^f^	100.1 ± 4.59 ^d^	3267.6 ± 149.74 ^e^	3960.7 ± 218.14 ^e^
NaVO		12.2 ± 0.25 ^b^	66.2 ± 2.32 ^b^	233.6 ± 11.68 ^b^	634.3 ± 44.40 ^c^
FOE		48.6 ± 2.85 ^d^	186.2 ± 7.45 ^g^	61.8 ± 3.73 ^a^	684.7 ± 27.39 ^cd^
Control		4.1 ± 0.17 ^a^	57.0 ± 3.42 ^a^	47.2 ± 3.35 ^a^	304.1 ± 13.71 ^b^
MeJA1	13	64.1 ± 2.59 ^b^	69.8 ± 4.58 ^b^	116.2 ± 2.92 ^a^	1579.7 ± 118.57 ^d^
cJA		n.d.	61.1 ± 2.44 ^a^	n.d.	143.2 ± 5.16 ^a^
2HJA		n.d.	87.6 ± 3.65 ^d^	n.d.	85.7 ± 0.49 ^a^
MeJA2		126.9 ± 2.64 ^c^	122.6 ± 3.75 ^f^	1236.9 ± 49.48 ^c^	2652.6 ± 136.12 ^e^
TvC1		169.9 ± 7.79 ^d^	70.3 ± 2.47 ^b^	2179.6 ± 132.58 ^d^	3518.2 ± 158.64 ^f^
TvC2		352.3 ± 7.05 ^e^	122.5 ± 2.12 ^e^	3226.6 ± 322.66 ^e^	4587.2 ± 199.95 ^g^
NaVO		65.6 ± 4.47 ^b^	127.0 ± 3.88 ^f^	667.9 ± 34.27 ^b^	1327.2 ± 53.09 ^c^
FOE		58.1 ± 2.01 ^b^	193.0 ± 6.78 ^g^	56.2 ± 2.58 ^a^	1189.1 ± 79.17 ^c^
Control		8.8 ± 0.18 ^a^	76.0 ± 1.91 ^c^	60.8 ± 4.86 ^a^	473.4 ± 17.07 ^b^

## Data Availability

Not applicable.

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
