# Peer review of "Biotic and Abiotic Elicitors of Stilbenes Production in *Vitis vinifera* L. Cell Culture"

_plants, 2021, doi:10.3390/plants10030490_

Round 1

Reviewer 1 Report

Stilbenes represent an interesting group of molecules among natural compounds. The number of references to this class of compounds has reached the tens of thousands for their interesting biological activity as antifungal compounds in plants and for their role in human health and disease. The manuscript “Biotic and Abiotic Elicitors of Stilbenes Production in Vitis  vinifera L. Cell Culture” deals with the competence of red-grapevine cell cultures to produce stilbenes under biotic and abiotic treatments. The authors show that maximum production of trans-resveratrol, δ- and ɛ-viniferins trigger by treatment with cellulase from T. viride. This study suggests that both used fungal elicitors can enhance production more effectively than commonly used jasmonates.

The manuscript is of interest and reports interesting new findings on the possible use of elicited cell cultures to obtain stilbenes for use in agriculture as natural pesticides as ecologically friendly bio-control agents for agriculture.

Minor revision are included in the notes of the munscript. 

The results are well presented only figure 1 needs to show the error bars and a statistical analysis.

Author Response

All our responses are in attached file.

Reviewer 2 Report

Dear authors, 

Line 153. and 318. I would prefer precisely explain why the measurement is made after 8, 11 and 13 days

Line 345. The information concerning pore size and particle size are
missing. Please add together with vendor and country of the origin of C8
sorbans.
Line 350. Please provide information of about HPLC-DAD model and vendor.
Line 360. Please change of mg mL-1 please to mg/mL.

Just comment: if we consider the grapevine as a specie, one cultivar is not enought for developing methodology.

Author Response

All our responses are in attached file and in text of manuscript.

Reviewer 3 Report

The article "Biotic and abiotic elicitors of stilbenes production in Vitis vinifera L. cell culture" describes the accumulation of stilbenes (trans-resveratrol, trans-piceid, δ-viniferin, and ɛ-viniferin) in Vitis vinifera L. cell culture after elicitation with biotic and abiotic elicitors (MeJA, 2HJA, cJA, FOE, TvC1, TvC2, and NaVO). The article is well written and presents interesting results for the production of stilbenes. I recommend to accept this article with minor corrections.

The results section should be amended to make it clearer and the comparison of results between different elicitors easier to understand. A choice must also be made to keep either Table 1 or Figures 3 and 4. They present the same results and this leads to confusion.

line 34-35: the sentence must be reformulated.

line 36: phytoalxin is not restricted to polyphenols.

line 56: delete "neither".

line 58-60: "Three in vitro cultivation systems, the cell suspension cultures, callus cultures, and hairy root cultures were..." must be replaced by "Three in vitro cultivation systems (the cell suspension cultures, callus cultures, and hairy root cultures) were..."

line 141: where are the results for cell viability ?

Figure 1: change "in vitro" by "in vitro" and "g L-1" by " g.L-1". Specify the code for elicitors in the legend.

line 152: change "in vitro" by "in vitro".

line 156: change"Trichoderma viride" by "Trichoderma viride".

line 163: change "Fusarium oxysporum" by "Fusarium oxysporum".

line 167: add "in" before Figure 3.

Figure 2: indicate the legend for the 4 windows (A, B, C, D). It will be better to change the gradient or column to allow faster elution of ε-viniferin and δ-viniferin.

line 181: change "in vitro" by "in vitro".

line 184: add "in" before Figure 4.

line 184-186: the sentence must be reformulated.

Table 1: Specify the unite of data. Indicate the number of replicates, the name and the significance threshold of the statistical test. In the table, start for each day with the control plants.

Figure 3 and 4: Indicate the number of replicates and add the results of the statistical test.

In the part " Elicitor preparation and elicitor treatment" add the concentration for elicitors.

line 355: change "min-1" by "min-1"

line 360: change " mg mL-1" by " mg.mL-1"

What do you want to do ? New mailCopy

Author Response

All our response sare in attached file and in revised manuscript.
